# Key Node Ranking in Complex Networks: A Novel Entropy and Mutual Information-Based Approach

**DOI:** 10.3390/e22010052

**Published:** 2019-12-30

**Authors:** Yichuan Li, Weihong Cai, Yao Li, Xin Du

**Affiliations:** 1Department of Computer Science, Shantou University, Shantou 515063, China18yli14@stu.edu.cn (Y.L.);; 2Key Laboratory of Intelligent Manufacturing Technology, Ministry of Education, Shantou University, Shantou 515063, China

**Keywords:** key nodes, complex network, entropy, mutual information

## Abstract

Numerous problems in many fields can be solved effectively through the approach of modeling by complex network analysis. Finding key nodes is one of the most important and challenging problems in network analysis. In previous studies, methods have been proposed to identify key nodes. However, they rely mainly on a limited field of local information, lack large-scale access to global information, and are also usually NP-hard. In this paper, a novel entropy and mutual information-based centrality approach (EMI) is proposed, which attempts to capture a far wider range and a greater abundance of information for assessing how vital a node is. We have developed countermeasures to assess the influence of nodes: EMI is no longer confined to neighbor nodes, and both topological and digital network characteristics are taken into account. We employ mutual information to fix a flaw that exists in many methods. Experiments on real-world connected networks demonstrate the outstanding performance of the proposed approach in both correctness and efficiency as compared with previous approaches.

## 1. Introduction

Complex network analysis has been used in many real-life networks, such as transportation networks, power networks, biological networks, and other complex systems [1,2]. Ranking the key nodes of complex networks is of great theoretical and practical significance for many applications. For example, if a key individual species is accurately confirmed, researchers claim, then key animal protection can assist with population renewal, sustainable production of a safe and varied food supply, and the eventual repair of ecosystems that have been damaged due to illegal hunting [3,4]. In the field of network security and cloud computing, research shows that the entire network will fall into paralysis when 5%–10% of all key nodes are attacked [5]. Protecting key nodes can not only improve the survivability and sturdiness of the entire network but also increase the network’s capacity to resist conventional attacks. The present study may also help to combat crime, in [6], a forensic analysis system called the ‘crime investigation system’, which uses the concept of ‘relative importance’, was shown to help forensic investigators determine the most influential members of a criminal group, who are related to the known members of the group, for the purposes of investigation. In this paper, we present a novel model based on the theory of entropy to facilitate the spread of information by choosing key nodes, which is of remarkable significance to many applications, such as floating rumors and epidemics control [7,8], restoring a power supply in a timely manner [9], and creating product promotion strategies [10].

To date, many methods have been proposed to identify key nodes in complex networks. The oldest and simplest method is degree centrality [11], which was proposed in 1979 by Freeman. This method measures the importance of nodes by counting how many neighbors a node has. Degree centrality has the merits of a simple calculation process and high calculation efficiency. However, it only uses local information and does not consider some other characteristics of the network. In fact, according to degree centrality, some nomadic nodes with many neighbors that are isolated from the main part of the network will be ranked at the top by mistake. To deal with this problem, Sabidussi and Chavdar proposed closeness centrality [12,13], which computes the mean distance that it takes a node to reach every other node in the network. This measure performs well in small-scale networks, but it is not suitable for processing large-scale networks because of its high computational complexity. To obtain a more efficient and accurate algorithm, Kitsak then proposed the K-shell algorithm [7]. In this method, every node is assigned a K-shell index. At each step, the K-shell index increases as the nodes in the outer layer are deleted from the remaining subnetwork. The core of a network will have the highest K-shell index and will be believed to be the most important node in the entire network. However, in practical terms, some nodes play a ‘ship and bridge’ role in the network; that is, communications between any two nodes have to pass through these nodes. Even though these nodes are not the core node, they still play a very important role. Therefore, betweenness centrality [14] was proposed to measure the importance of a node by counting the number of times that a node is located at the shortest path between any two other nodes. In addition, many excellent and effective methods, such as eigenvector centrality [15], stress centrality [16], and eccentricity centrality [17], have been proposed and widely used. However, the above-discussed methods often come into conflict with each other; for example, a node may have high a centrality value in one method but a lower value in another [18]. As can be seen in Figure 1, even in the same network, the centrality values that are obtained by these methods have nothing in common with each other [19]. To be sure, all of these methods have some limitations and specific application scenarios that are related to the way they consider the problem. A valid method for ranking nodes in a complex network remains an open issue.

Recently, researchers have been attempting to solve this problem by using information entropy. Some excellent entropy-based methods have been proposed, such as entropy variation [20] and subgraph-entropy-based centrality [21]. Subgraph-entropy-based centrality was proposed in order to identify important nodes in weighted networks. It takes the weight into account, which is of great significance as it makes the result more accurate. However, this method only makes use of the neighbors of a node and is used in a weighted network. Entropy variation is one of the systems analysis methods that applies entropy to find key nodes. However, it only utilizes local information to estimate the importance of nodes and only has application in directed networks. Some excellent entropic indexes have been proposed and used to obtain some results in ranking key nodes. How can we take full advantage of information entropy? We need to continue to explore.

In this paper, to provide a more efficient and extensible framework, we propose a novel entropy and mutual information-based measure (EMI). A given complex network may have multiple morphological features or its morphological features may not be completely clear, as a single index may not interpret some morphological features [22]. In our method, we consider as many features as possible. For weighted networks, we rank the nodes in two respects: structural features and parameter characteristics. Here, structural features are no longer restricted to neighbors of a node; for directed networks, we consider a greater range of information and the directions of the edges. In particular, our algorithm has very low time complexity even though a wider range of features are involved in the computation. Furthermore, we found that nodes can be overestimated by existing approaches, and the overestimation is worse when the network consists of a large number of nodes with a dense distribution. For example, consider node i and node j, two adjacent nodes, that have many neighbors in common. If we have already offered node i a high ranking, then node j is not more important than node k who has less followers than node j but is completely different from nodes i and j. To clear up this drawback, we propose a method based on mutual information that can determine to what extent a node is overvalued. Since plenty of possible factors are taken into account, EMI obtains outstanding performance.

The main contribution of this article can be stated as follows. First, we present a novel entropy and mutual information-based algorithm. Second, EMI is the first method that can be applied to all kinds of networks and obtain excellent results. Third, features in the network are fully captured and the range of information that we adopt is wider than ever before, which makes the ranking more accurate. Furthermore, we are the first to raise the issue of overvaluation and settle this question by utilizing mutual information theory. Last but not least, the time complexity of our algorithm is far lower than that of the existing methods.

The remainder of this paper is organized as follows. In Section 2, to help readers quickly understand this field, we provide some required background knowledge and describe some well-known centrality methods as a quick guide. In Section 3, some definitions are given and our method is introduced and explained with a simple case in detail. Datasets and evaluation criteria are reported in Section 4. In Section 4, we also describe six laboratory experiments that were done, and compare the effectiveness and performance of EMI with that of some other famous methods. Finally, we conclude the paper in Section 5.

## 2. Background

In this section, some famous and widely used centrality methods are introduced in detail. We also choose some of them to be the benchmark algorithms for EMI. The degree centrality method and the K-shell decomposition algorithm are classical local approaches that are based on neighbors. Betweenness centrality and closeness centrality are representatives of the class of path-based methods. Google’s famous page ranking algorithm, PageRank, is a well-known random walk method, which also is one of the Eigenvector centrality methods. The susceptible-infected (SI) model is also introduced in this section. It is employed to calculate the proposed model’s identification accuracy and the benchmark methods.

### 2.1. Degree Centrality

The historically first and conceptually simplest method is degree centrality, which identifies the importance of a node by the number of links incident upon a node. The degree can be regarded as the communication exchange center that attracts any other node converges on it [23]. In a directed network where ties have direction, degree centrality defines two separate indexes, namely the indegree and the outdegree. Accordingly, the indegree is the number of nodes that point to the node and the outdegree is the number of ties that the node directs to others. The degree centrality *d_c_* of a vertex v, for a given graph, is defined as:(1)$$d_{c}\left(v\right)=\sum_{i=1,i\neq j}^{n}t_{ij},$$
where *n* is the total number of peers in the network, j stands for the operating node v, i represents the rest of the nodes in the graph, and $t_{ij}$ is the number of ties between j and i, which equals 1 if a tie exists or 0 otherwise.

### 2.2. Closeness Centrality

In a connected network, the standard closeness centrality of a node is defined as the average length of the shortest path between the node and all other nodes in the graph. In this model, the more centrally located a node is, the closer it is to all other nodes. Closeness centrality was proposed by Alex Bavelas in 1950 as the reciprocal of the farness [24], that is:(2)$$C\left(x\right)=\frac{1}{\sum_{y}d\left(x,y\right)},$$
where d(x,y) expresses the distance between peers x and y. Taking distances from or to all other nodes is not just relevant to path in weighted graphs. It can produce totally different results in weighted graphs, for example, distances between pairs of neighbor nodes may be different.

### 2.3. K-Shell

The K-shell algorithm determines the importance of nodes based on their topological locations. It defines the concept of a k-core as a subnetwork, which is obtained by removing the nodes until the degree of the nodes in the subgraph is not less than k. The k-core algorithm first defines the nodes whose degree is 1 as 1-core and deletes the connections of these nodes from the network. Then, the algorithm continues identifying the nodes whose degree is 1 as 2-cores, and removes the connections of these nodes. By repeating this step, the method finds the core of the network, which is the node with the largest k-core value [7].

### 2.4. Betweenness Centrality

Betweenness centrality takes both the global features and the disconnected components into account to quantify the number of times a node acts as a bridge along the shortest path between two other nodes. It was reported as a method for measuring the control of a human over the communication between other humans in a social network by Linton Freeman in 1977 [14]. The betweenness of a vertex *v* can be computed using the following steps:For each pair of vertices in the graph, compute the shortest paths between them.For each pair of vertices, count the number of shortest paths that pass through node *v*.Sum this number over all pairs of vertices.

More compactly, in a connected graph, betweenness can be represented as:(3)$$C_{B}\left(v\right)=\sum_{s\neq v\neq t\in V}\frac{\sigma_{st}\left(v\right)}{\sigma_{st}},$$
where $\sigma_{st}$ is the total number of shortest paths from node s to node t and $\sigma_{st}\left(v\right)$ is the number of those paths that pass through v. Betweenness is commonly normalized by dividing the number of pairs of vertices excluding v, which for directed networks is (n−1)(n−2) and for undirected networks is (n−1)(n−2)/2. Thus, the normalized betweenness of node *v* is computed as:(4)$$C_{NB}\left(v\right)=\frac{2}{\left(N-1\right)\left(N-2\right)}\underset{s\neq v\neq t\in V}{\sum }\frac{\sigma_{st}\left(v\right)}{\sigma_{st}}.$$

### 2.5. Eigenvector Centrality

Eigenvector centrality [25] assigns voting scores to all nodes in the network based on the idea that connections to high-scoring nodes contribute more to the score of the node than connections to low-scoring nodes. Google’s PageRank [26] and Katz centrality [27] are based on the same idea. In Eigenvector centrality, we often use the adjacency matrix to obtain the centrality score. Given a network, let $A=\left(a_{v,t}\right)$ be the adjacency matrix. $a_{v,t}=1$ if vertex v is linked to vertex t, and $a_{v,t}=0$ otherwise. The voting centrality score of node v can be defined as:(5)$$x_{v}=\frac{1}{\lambda }\sum_{t\in M\left(v\right)}x_{t}=\frac{1}{\lambda }\sum_{t\in G}a_{v,t}x_{t},$$
where M(v) is a set of the neighbors of v and λ is a constant. With a small rearrangement, this can be rewritten in vector notation as the eigenvector equation Ax=λx [15].

PageRank, which is used by Google Search to rank web pages in their search engine results, is an algorithm that is the most representative method based on Eigenvector centrality. According to Google, PageRank works by counting the number and quality of links to a page to obtain a rough estimate of how important the website is. The underlying assumption is that more important websites are likely to receive more links from other websites. The PageRank value of a webpage is calculated by two main steps. Firstly, the algorithm assigns every page an initial value adding up to 1. Secondly, the PageRank values of every page are updated with:(6)$$PR\left(i\right)=A^{T}\times PR\left(i-1\right),$$
where PR(i) is the matrix of the PageRank values in the ith iteration and PR(i−1) is the matrix of the PageRank values in the i−1 iteration.

### 2.6. Susceptible-Infected Model

Susceptible-infected (SI) model is one of the most popular epidemic spreading models for analyzing spreading processes such as infectious disease, rumors, and information. It is also one of the simplest compartmental models. Many models are derivations of this basic form [28]. The model is reasonably predictive of infectious diseases that are transmitted from human to human. All crowds are initially classified into two categories: S for the number of susceptible people, and I for the number of infectious people. These variables (S, I) represent the number of people in each compartment at a particular time. To represent that the number of susceptible and infected individuals may vary over time, even if the total population size remains constant, we make the precise numbers a function of t(time): S(t), I(t). For a specific disease in a specific population, these functions may be applied to predict possible outbreaks and bring them under control. The classical model SI also includes another category of people: R, which stands for people who have been recovered and will never be infected again. In this model, the infected people spread the disease and infect their neighbors with a probability of γ and then become the recovered state R, where they are immunized and cannot be infected again. The model with state R is suitable for the complicated propagation processes. In this paper, to efficiently examine the identification accuracy of EMI and other methods, we employed the SI model without state R to do this. The total number of infected peers could be considered as an indicator to evaluate the influence of the initially infected node. 

## 3. Materials and Methods

### 3.1. Preliminaries

Given a directed, weighted network G=(V,E,W), where $V=\left\{v_{1},\ldots ,v_{\left|V\right|}\right\}$ denotes the set of vertices, and $E=\left\{e_{1},\ldots ,e_{\left|E\right|}\right\}$ is the set of directed edges that often describes the relations between the nodes, the set $W=\left\{w_{1},\ldots ,w_{\left|w\right|}\right\}$ corresponds to the weight on the edges. The number of nodes, edges, and weights are expressed as |V|, |E|, and |W|, respectively. The number of edges that point to others of a node is called the outdegree; similarly, the number of inbound edges of a node is called its indegree. Nodes $v_{i}$ and $v_{j}$ are called neighbors to each other if there is an edge between them. In this paper, we defined node $v_{k}$, who is the neighbor of $v_{j}$ but not that of $v_{i}$, to be the secondary neighbor of $v_{i}$. Since the secondary neighbors of a node are connected directly to its neighbors, the number of one’s secondary neighbors can be obtained by counting the degrees of all its neighbors, then subtracting the degree itself. Inspired by the data structure discipline, a node with a lot of secondary neighbors is regarded as being just like the root node of a full binary tree, which means that more and more nodes can be accessed from this node. The node can be regarded as the core of an incremental neighbor network, and communications or information transmissions from this node can reach far and wide. It possesses effective power over the nodes that are connected to it directly and mediately. Motivated by the above discussions, we supposed that the topological influence of a node could be obtained by employing the secondary neighbors properly.

The weights carried by the edge can also describe the importance or influence of a node. In a trust network, for example, the weight between two users represents the degree of trust of one user to the other. If one user is trusted deeply by some users with great weights received from other users, then we believed that this user greatly influences other users. Analogously, weights were used to assess interactive power. Based on these preliminaries, we presented our method for assessing the power of each node via degree and weight-based entropy centralities.

### 3.2. Preliminaries for Information Entropy

We begin this subsection with the definition of information entropy. Information entropy is the average rate at which information is produced by a stochastic source of data [29]. The measure of the information entropy that is associated with each possible data value is the negative logarithm of the probability mass function for the value. If *E* is a set of possible events $e_{1}\;\ldots \;e_{n}$, and $p_{i}$ is the probability that event $e_{i}$ occurs, and $\sum_{i=1}^{n}p_{i}=1$, the entropy of *E* is calculated by:(7)$$H=-\overset{n}{\underset{i=1}{\sum }}p_{i}*\log\left(p_{i}\right).$$

When the data source produces a low probability value, in other words, a low-probability event occurs, this low-probability event carries more information than when the source data produces a high probability value. The amount of information that is conveyed by each event, when defined in this way, becomes a random variable whose expected value is the information entropy. Intuitively speaking, entropy refers to disorder and uncertainty; the definition of entropy that is used in information theory is directly analogous to the definition that is used in statistical thermodynamics. The concept of information entropy was introduced by Claude Shannon in 1948 [30]. The basic idea of information theory is that the new event of a communicated message depends on the degree to which the content of the message is surprising. If an event is very probable, it is no surprise when the event happens as expected, and it has nothing to do with the system as a whole. Here, we measured the surprise and importance of a node to the network by modeling information entropy.

**Definition** **1.**
*Let N = ($v_{1}$, $v_{2}$, …, $v_{n}$) be a set of nonzero positive integers. This set forms a probability distribution p = ($p_{1}$, $p_{2}$, …, $p_{n}$), in which:*
(8)$$p_{i}=\frac{v_{i}}{\sum_{k=1}^{n}v_{k}},\;i=1,\;2,\;\ldots ,n.$$


Therefore, the entropy of the set *N* = ($v_{1}$, $v_{2}$, …, $v_{n}$) is calculated by:(9)$$H\left(v_{1},\;v_{2},\;\ldots ,\;v_{n}\right)=-\overset{n}{\underset{i=1}{\sum }}\frac{v_{i}}{\sum_{k=1}^{n}v_{k}}log\left(\frac{v_{i}}{\sum_{k=1}^{n}v_{k}}\right).$$

**Definition** **2.**
*For a node*
$v_{i}$
*in the network, a characteristic manipulation function f(v_i_) can be the function that is used to calculate the degree or the weight. We define:*
(10)$$p\left(v_{i}\right)=\frac{f(v_{i})}{\sum_{k=1}^{n}f\left(v_{k}\right)}.$$


Since 0 ≤
*p*(*v_i_*) ≤ 1, it can be regarded as a probability.

**Definition** **3.**
*The entropy of node*
$v_{i}$
*with the characteristic manipulation function f is defined as:*
(11)$$H=-\overset{n}{\underset{i=1}{\sum }}\frac{f(v_{i})}{\sum_{k=1}^{n}f\left(v_{k}\right)}\log\frac{f(v_{i})}{\sum_{k=1}^{n}f\left(v_{k}\right)}.$$


### 3.3. Description of the Method

In a directed and weighted network, we separated structural entropy centrality into two parts: indegree entropy centrality and outdegree entropy centrality. These two centralities can be very helpful to identify key nodes. A simple example to illustrate this state is as follows: suppose that there is a website network. If page A follows lots of pages, there will be many links from page A to other pages. If there are thousands of pages that cite page B, then page B will receive a high inbound degree. Thus, the outbound degree of a node refers to its vitality and the inbound degree of a node interprets its authority and popularity. Based on the above idea, the following definitions of structural entropy centrality are given.

The indegree of node $v_{i}$ is denoted by $d_{i}^{in}$ and, correspondingly, $d_{i}^{out}$ is the outdegree of node $v_{i}$. $N_{i}=\left(v_{1},\;v_{2},\;\ldots ,\;v_{n}\right)$ represents the set of all the neighbors of *v_i_*. The total indegree and outdegree of $N_{i}$ are defined as $EXD_{i}^{in}$ and $EXD_{i}^{out}$, respectively. They are calculated as follows:(12)$$EXD_{i}^{in}=\sum_{j=1}^{n}d_{j}^{in},$$
(13)$$EXD_{i}^{out}=\sum_{j=1}^{n}d_{j}^{out},$$
where, $d_{j}^{in}$ and $EXD_{i}^{in}$ originate from Definitions 1 and 2. In the formula, the characteristic manipulation function computes the indegree of node $v_{j}$. $d_{j}^{in}$ equates to $f(v_{j})$ and $EXD_{i}^{in}$ is equivalent to $\overset{n}{\underset{k=1}{\sum }}f\left(v_{k}\right)$. Based on Definition 3, the indegree and outdegree entropy centralities, namely $S_{i}^{in}$ and $S_{i}^{out}$, of node $v_{i}$ are calculated by the following formulas:(14)$$S_{i}^{in}=-\sum_{j=1}^{n}\frac{d_{j}^{in}}{EXD_{i}^{in}}log\frac{d_{j}^{in}}{EXD_{i}^{in}}.$$
(15)$$S_{i}^{out}=-\overset{n}{\underset{j=1}{\sum }}\frac{d_{j}^{out}}{EXD_{i}^{out}}log\frac{d_{j}^{out}}{EXD_{i}^{out}}.$$

The final structural entropy centrality $S_{i}$ is:(16)$$S_{i}=S_{i}^{in}+S_{i}^{out}.$$

Notably, in a complex network, there are other characteristics than structural features that describe the importance of a node [31]. For example, in an airport network, each edge represents an air route from one airport to another, and the weight of an edge shows the number of flights on that route in the given direction. In addition, the weight of an edge often describes the trust, strength, popularity, and willingness of a node in a particular network. Therefore, we proposed that weights on the connection of two nodes could reflect the status of nodes and weight-based entropy centralities as follows.

Node $v_{j}$ is one of the neighbors of $v_{i}$. $w_{j}^{in}$ is defined as the sum of the weights on the inbound edges of node $v_{j}$. Similarly, $w_{j}^{out}$ is defined as the sum of the weights on the outbound edges. The total weights of the indegree and the outdegree of $v_{i}$’s neighbors are computed by:(17)$$EXW_{i}^{in}=\sum_{j=1}^{n}w_{j}^{in}.$$
(18)$$EXW_{i}^{out}=\sum_{j=1}^{n}w_{j}^{out}.$$

Then, the components of interactive entropy can be obtained by:(19)$$I_{i}^{in}=-\sum_{j=1}^{n}\frac{w_{j}^{in}}{EXW_{i}^{in}}log\frac{w_{j}^{in}}{EXW_{i}^{in}}.$$
(20)$$I_{i}^{out}=-\overset{n}{\underset{j=1}{\sum }}\frac{w_{j}^{out}}{EXW_{i}^{out}}log\frac{w_{j}^{out}}{EXW_{i}^{out}}.$$

Naturally, the value that is measured by interactive entropy is:(21)$$I_{i}=I_{i}^{in}+I_{i}^{out}.$$

As was discussed above, using traditional methods usually causes a redundant influence to occur. In other words, nodes’ influence may be overrated in a dense network. In this paper, we employed mutual information to solve this problem. Intuitively, mutual information measures the information that events X and Y share. It measures the degree to which knowing one of these variables reduces uncertainty about the other [32]. For example, if X and Y are independent, then knowing X does not provide any information about Y; that is, their mutual information is zero. At the other extreme, if X is a deterministic function of Y and Y is a deterministic function of X, then all information conveyed by X is shared with Y, which means that knowing X determines the value of Y and vice versa. As shown in Figure 2, X and Y are two correlated variables. The circle on the left side is the individual entropy H(X) with the red area being the conditional entropy H(X|Y). The circle on the right side is the individual entropy H(Y), with the blue part being the conditional entropy H(Y|X). The area that is covered by both circles is the joint entropy H(X,Y). The middle part that is colored violet is mutual information I(X;Y) [33]. If the two circles stand for the area of influence of two connected nodes X and Y, it will be reasonable to measure the mutual power (the violet part) by using mutual information.

Motivated by this, and based on mutual information theory, the mutual power of two connected nodes is defined as follows:

**Definition** **4.**
*We define two terms,*
$N_{i}$
*and*
$N_{j}$
*, which are the set of all the neighbors of two connected nodes*
$v_{i}$
*and*
$v_{j}$
*, respectively.*
$N_{ij}$
*is defined as the set of the common nodes of*
$N_{i}$
*and*
Nj
*.*
$N_{\left(i,j\right)}$
*is defined as the joint set of*
$N_{i}$
*and*
$N_{j}$
*, which equals the combination of*
$N_{i}$
*and*
$N_{j}$
*.*


**Definition** **5.**
*Given sets $N_{i}$ and $N_{j}$, we define the random events X and Y by picking one node at random from each set that satisfies $X:\left\{x\in N_{ij}|x\in N_{i}\right\}$ or $Y:\left\{y\in N_{ij}|y\in N_{j}\right\}$. The size function F(X) is defined in order to calculate the size of the set.*


Therefore, the joint probability distribution of the random events X and Y is $F\left(N_{ij}\right)/F\left(N_{\left(i,j\right)}\right)$, and the marginal probability distribution of X and Y is $F\left(N_{ij}\right)/F\left(N_{i}\right)$ and $F\left(N_{ij}\right)/F\left(N_{j}\right)$, respectively. In the mutual information theory, the mutual information is equal to zero when the joint distribution coincides with the product of the two marginals, which means that X and Y are independent. For a given graph, two connected nodes are independent if they have no common neighbors. However, in real-life complex networks, two connected nodes with few common neighbors is a pervasive phenomenon. In this context, two connected nodes are considered to be independent when they only have very few common neighbors, and the joint probability distribution and the product of the marginals should be nearly the same. Consequently, we set the product of the marginals to $F\left(N_{ij}\right)/\left(F\left(N_{i}\right)+F\left(N_{j}\right)-F\left(N_{ij}\right)\right)$, which will be very close to the joint distribution when the number of common neighbors is very low. Another reason for this is that if $F^{2}\left(N_{ij}\right)/\left(F\left(N_{i}\right)\cdot F\left(N_{j}\right)\right)$ is used as the product of the marginals, the denominator will be zero in the case of $F\left(N_{ij}\right)=0$. According to the above description and discussion, the mutual influence between node $v_{i}$ and its neighbors can be obtained based on the mutual information theory as the following formula: (22)$$MI_{i}=\underset{x\in X}{\sum }\underset{y\in Y}{\sum }\frac{F\left(N_{ij}\right)}{F\left(N_{\left(i,j\right)}\right)}log\frac{F\left(N_{i}\right)+F\left(N_{j}\right)-F\left(N_{ij}\right)}{F\left(N_{\left(i,j\right)}\right)}.$$

The eventual influence value of node $v_{i}$ is:(23)$$K_{i}=S_{i}+I_{i}+MI_{i}.$$

For other kinds of network, if the network has either directions or weights, then the indegree and outdegree entropy centralities or the interactive entropy are computed for it. For undirected and unweighted complex networks, we only need to compute the structural entropy centrality, which is defined in the following formula. The mutual power is considered for all kinds of complex networks.
(24)$$S_{i}=-\overset{n}{\underset{j=1}{\sum }}\frac{d_{j}}{EXD_{i}}log\frac{d_{j}}{EXD_{i}}.$$

The final influence value of the node is:(25)$$K_{i}=S_{i}+MI_{i}.$$

Notice that in the subgraph method [34], the authors also employ the neighbors’ inbound degree and outbound degree, but what is different is that not all of the degrees of the neighbors are used. For example, let us assume that there is a subgraph centered on node ***A*** who has neighbors including nodes ***B***, ***C***, and ***D***. In the subgraph method’s process of calculation, the authors only take the edges that are connected to node ***A*** into account. In our proposed method, however, we make use of all of the edges of neighbors, including edges that point to other nodes. In Ahmad Zareie’s research, the authors make use of all of the degrees of the secondary neighbors, which leads to high computational complexity [35]. Compared with these two methods, our approach seems to be a compromise: EMI considers global features and has low computational complexity. Our experiments show that EMI takes advantage of both accuracies and efficiencies. A clear and standardized pseudo-code of the proposed method is elaborated in Algorithm 1.

According to Algorithm 1, the network type is first detected. Then, based on the features, the different entropy centralities of each node are calculated. Finally, we ranked all of the nodes in descending order according to the $K_{i}$ values and obtain the top-k key nodes from the order list.
**Algorithm 1.** EMI. **Input:**   G=(V,E,W) with n=|V|, m=|E|, q=|W|**Output:**   Ranked nodes with overall importance value**Begin algorithm**1:  **If**
*G* is directed or weighted:2:  **For** node $v_{i}$ in V3:    compute structural entropy centrality $S_{i}^{in}$, $S_{i}^{out}$, and $S_{i}$4:    compute interactive entropy centrality $I_{i}^{in}$, $I_{i}^{out}$, and $I_{i}$5:    compute mutual power $MI_{i}$6:    compute overall importance $K_{i}$7:  **Else**8:    **For** node $v_{i}$ in V9:        compute structural entropy centrality $S_{i}$10:      compute mutual power $MI_{i}$11:      compute overall importance $K_{i}$12:  **End for**13:  rank nodes based on key values**End algorithm**

### 3.4. Case Study

To make it easy to understand how to use Algorithm 1 to assess the value of nodes’ importance, we described the whole process in detail by making the following simple graph, shown in Figure 3, an example [36]:

We used node *a* as an example and a complete ranking list will be given in the end of this subsection. Some parameters that were used in the process of calculation are listed in Table 1.

In this paper, we set 10 as the base value. Then, the structural entropy, interactive entropy, and overall power could be obtained by the algorithm as follows.
(26)$$S_{a}=S_{a}^{in}+S_{a}^{out}=-\left(\sum_{j=1}^{n}\frac{d_{j}^{in}}{EXD_{a}^{in}}log\frac{d_{j}^{in}}{EXD_{a}^{in}}+\sum_{j=1}^{n}\frac{d_{j}^{out}}{EXD_{a}^{out}}log\frac{d_{j}^{out}}{EXD_{a}^{out}}\right)=0.7782.$$
(27)$$I_{a}=I_{a}^{in}+I_{a}^{out}=-\left(\sum_{j=1}^{n}\frac{w_{j}^{in}}{EXW_{a}^{in}}log\frac{w_{j}^{in}}{EXW_{a}^{in}}+\sum_{j=1}^{n}\frac{w_{j}^{out}}{EXW_{a}^{out}}log\frac{w_{j}^{out}}{EXW_{a}^{out}}\right)=0.7073.$$
(28)$$MI_{a}=\sum_{x\in X}\sum_{y\in Y}\frac{F\left(N_{aj}\right)}{F\left(N_{\left(a,j\right)}\right)}log\frac{F\left(N_{a}\right)+F\left(N_{j}\right)-F\left(N_{aj}\right)}{F\left(N_{\left(a,j\right)}\right)}=-0.03876.$$
(29)$$K_{a}=S_{a}+I_{a}+MI_{a}=1.44674.$$

As the above illustration shows, the overall power of each node can be obtained in the same way. The results are listed in Table 2. According to the value of the overall power of each node, the ranking list is illustrated in Table 3.

## 4. Experiments and Results

Details on the experiments, including the selected datasets, the evaluation criteria, and the performance of all of the measures, were reported in this section. To obtain a convincing evaluation of EMI, we compared it with the following measures:Degree centrality (Deg) [11];K-shell (K-s) [7];Closeness centrality (Clo) [12];Betweenness centrality (Bet) [14];Stress centrality (Str) [16];Radiality centrality (Rad) [37];Bridging centrality (Bri) [38];Centroid centrality (Cen) [39];Eccentricity centrality (Ecc) [17]; andDensity of maximum neighborhood component (DMNC) [40,41];

For this purpose, the codes were implemented in Python 3.7 and experiments were performed on a computer with an Intel Core i5 2.4 GHz processor and 8 GB RAM.

### 4.1. Datasets

We applied our proposed ranking measure to six real networks to evaluate its performance. To test the pervasiveness of EMI, these datasets included different kinds of networks, including a weighted and directed network, an unweighted and undirected network, and a directed and unweighted network. All of these networks can be downloaded from http://konect.uni-koblenz.de/networks/. They are as follows:Dutch college (Dc): this directed and weighted network contains friendship ratings between 32 university freshmen who had never met before. Each of them was asked to rate the other students at different time points. The origin of the timestamps is not accurately known, but the distance between two timestamps is the same. Each node represents a student, and an edge between two nodes shows that one student rated another student. The edge weights show how good their friendship is from the perspective of the rating student. The weight ranges from −1 (a risk of getting into conflict) to +3 (best friend) [42].US-Airports (US-Air): this is a directed and weighted network of flights between U.S. airports in 2010. Each edge represents a flight path from one airport to another, and the weight of an edge shows the number of flights on that connection in the given direction in 2010 [43].Air traffic control (A-tc): this network is directed and unweighted. It was constructed from the United States’ FAA (Federal Aviation Administration) National Flight Data Center (NFDC) Preferred Routes Database. Nodes in this network represent airports or service centers and edges are created from strings of preferred routes that are recommended by the NFDC [44].Euroroad (E-road): this is the international E-road network, which is a road network that is located mostly in Europe and is undirected and unweighted. Nodes represent cities and an edge between two nodes denotes that they are connected by an E-road [45].Chicago: this is the road transportation network of the Chicago region in the United States; it is undirected and unweighted. Nodes are regarded as transports, and edges are connections [46].Dolphins: this directed social network was created by bottlenose dolphins. The nodes represent the bottlenose dolphins (genus *Tursiops*) of a bottlenose dolphin community living near Doubtful Sound, which is a fjord in New Zealand (spelled fiord in New Zealand). An edge indicates a frequent association. The dolphins were observed between 1994 and 2001 by oceanographers [47].

Figure 4 shows the layout of the six datasets. And Table 4 shows the basic topological features of the giant connected component of these networks, including the network size *n*, the number of edges *e*, the average degree *ad*, the maximum degree *d_max_*, the clustering coefficient *cc* [1], the assortativity ρ [48], and the relative edge distribution entropy σ.

### 4.2. Evaluation Criteria 

In this experiment, the monotonicity relation [49], as defined in Equation (30), was employed to evaluate the discriminability of a ranking measure. A ranking algorithm will be better if a few nodes are listed in the same rank.
(30)$$M\left(R\right)={\left(1-\frac{\sum_{r\in R}n_{r}\cdot \left(n_{r}-1\right)}{n\cdot \left(n-1\right)}\right)}^{2},$$
where n is the number of different ranks in a ranking list *R*, and $n_{r}$ is the number of nodes that have been listed in the same rank *r.* Obviously, if all of the nodes were to be placed in the same rank, the value of M(R) would be 0, and the result would be of little or no value in determining how important a node is. When all of the nodes receive a unique ranking, the value of M(R) would be 1, and the ranking result would thus be perfectly monotonic.

Moreover, the complementary cumulative distribution function (CCDF) was utilized [50], in addition to monotonicity, for a better evaluation of the ranking distribution of different methods. The value of the function is calculated for rank *r* by Equation (31):(31)$$CCDF\left(r\right)=\frac{\left|V\right|-\sum_{i=1}^{r}n_{i}}{\left|V\right|},$$
where $n_{i}$ represents the number of nodes that were placed in rank *i,* and |V| is the total number of network nodes. This function can better reflect the distribution of nodes to different ranks. Having more nodes gathered in a rank causes the function to sharply drop to zero, while having an extremely scattered distribution of nodes to different ranks makes the function have a mild slope. However, differentiation and monotonicity alone cannot be used to identify whether or not a ranking method is successful; hence, we have also tested the method in terms of its precision and the correctness of the resulting rankings.

In order to assess the precision of EMI, we compared the ranking list that was generated by EMI with those of the benchmarking methods using the real spreading influence model susceptible-infected (*SI*) [28]. In this model, each node belongs to one of two states: susceptible (*S*) or infectious (*I*). At first, all nodes are set to be in the state *S* except for node $v_{i}$, which is selected to be the infected node. At each timestamp, the infected nodes infect their susceptible neighbors with an infection probability β. The number of infected nodes is regarded as the influence of node $v_{i}$ when the epidemic process is finished. However, the infection probability is not constant in a weighted network. The feature that is contained on the directed edge has to be considered as well. According to Yan et al. [51], the infection probability in directed and weighted networks is defined as $\lambda_{ij}={\left(\frac{w_{ij}}{w_{max}}\right)}^{\alpha }$, in which susceptible node *i* is infected through neighbor node *j*. *α* corresponds to a constant of a positive value, $w_{ij}$ is the weight of a directed boundary, and $w_{max}$ denotes the maximum value among $w_{ij}$. In this experiment, we adopted $\lambda_{ij}$ as the infection probability in directed and weighted networks. We first used the EMI and other methods that were mentioned above to find the ten most critical nodes. Then, we selected different sets (different numbers of nodes, usually at the top of ranking list) as the seed nodes and made them the infected nodes. The number of infected nodes was regarded as the correctness of a method when the propagation process was over. To increase accuracy, we ran the process hundreds of times, and the mean value was considered to be the final result. Finally, time-efficiency is the last but not the least criteria that we need to evaluate. A shorter running time means that the program is fast and efficient.

### 4.3. Experimental Results

In the first experiment, other well-known centrality measures were applied to six real networks. The monotonicity value of the ranking list that was obtained using each of the approaches was calculated. Table 5 shows the results. The greater the value of M is, the more clearly a method can distinguish which rank the node belongs to. Based on the theory and Table 5, EMI obtained the maximum monotonicity values for all datasets except US-Air and Chicago; for these datasets, closeness centrality and bridging centrality obtained the greatest values (0.98480 and 0.93045, respectively), which were slightly higher than those of EMI. However, some methods performed poorly; for example, centroid, DMNC, and eccentricity performed poorly in most of the datasets, and even obtained a value of zero in particular datasets. It is remarkable that half of the methods had an average M value of less than 0.5. Plainly, EMI had outstanding performance as compared to these authoritative methods. The M values of EMI were vastly superior to those of most methods.

To further assess the efficiency of different measures in assigning distinct ranks to each node, we examined the node distribution in the next part of our experiment. To make the result more concise, we compared EMI with the top four measures (betweenness centrality, closeness centrality, bridging centrality, and stress centrality). Four datasets (Dutch college, US-Airport, Air traffic control, and Dolphins) were used in this part. As shown in Figure 5, EMI did a perfect job in Dutch college and Dolphins, as every node was assigned to a distinct rank. Closeness centrality did not behaved very well in these two datasets. In US-Airport, EMI and closeness centrality performed the best compared with stress centrality and betweenness centrality, which placed a lot of nodes in the same rank. In the Air traffic control network, although EMI placed dozens of nodes in the same rank, the number was far lower than those of the other four methods. As shown in Figure 5, stress centrality, betweenness centrality, and bridging centrality assigned almost 200 nodes to the same rank in the Air traffic control network. The ranking list that was obtained by closeness centrality had three ranks, into which were assigned 50, 100, and 150 nodes. Generally speaking, the ranking list that was obtained by EMI had a lower number of nodes in the different ranks, which is an urgent need when faced with networks containing millions of vertices.

In addition, the CCDF is plotted for Dutch college, US-Airport, E-road, and Dolphins in Figure 6. Due to the large number of data points, we still chose, as we did before, to compare EMI with the top four measures. As mentioned earlier, the lower the number of nodes in the same rank, the slower will the curve decrease to zero. In Dutch college and Dolphins, EMI performed as well as most of the well-known measures and much better than closeness centrality. From the US-Airport and E-road datasets, we could see that EMI emerged as the best candidate as the size of the network increases. Closeness centrality and stress centrality simply indicate that a node was heavily involved in the core but was not relevant to maintaining communication between other nodes. Accordingly, bridging centrality and betweenness centrality focus on the shortest path, could not differentiate between nodes with different influences, and the nodes were distributed in less-distinct ranks. EMI outperformed the top four well-known methods; thus, it could be seen that deep disparities remain between EMI and the other 10 famous methods. Based on the above conclusion, it could be inferred that EMI distributed nodes to a larger number of ranks, and each rank contained a lower number of nodes. EMI was the best choice when a large dataset requires ranking. In the next experiment, we assessed the correctness and accuracy of all of the measures.

Experiments were performed to examine the methods’ accuracy by comparing the top 10 vital nodes. Table 6 shows the lists of the top 10 key nodes produced by EMI and the other 10 approaches. In the Dutch college network, the proposed method, Bridging centrality, and DMNC had two and three nodes (red mark) in common. EMI is dissimilar to the other methods in terms of common nodes; however, five nodes is not a small number for a network with 32 vertices. In the US-Airport network, the methods almost have no resemblance to each other; EMI and DMNC were the only methods with half of the nodes in common. For the Air Traffic Control network, the number of the same nodes in the list between the proposed method and degree centrality, K-shell, closeness centrality, betweenness centrality, stress centrality, and radiality centrality was 8, 1, 6, 2, 4, and 6, respectively. EMI and four methods (closeness centrality, betweenness centrality, stress centrality, and radiality centrality) had only one node in common in the E-road network. In addition, the fact that the same nine nodes were identified by EMI and degree centrality is noteworthy, as it means that our method performs as well as degree centrality. Moreover, it can be seen that, in the Chicago and Dolphins networks, our proposed method and the other popular methods have almost all of the nodes in common. To sum up, we could observe that our proposed measure could produce acceptable results and provided accurate ranks for the top-N nodes in comparison with other methods. There was no conflict between EMI and other well-known methods.

As the datasets were ranked by all of the measures, we ran the SI simulation on the six datasets. In light of the size of the networks, we selected the top 2, 4, 6, 8, and 10 nodes to be the seed set in the Dutch college and Dolphins networks, and the top 10, 20, 30, 40, and 50 nodes to be the seed set in the US-Airport, Air traffic control, E-road, and Chicago networks. Correspondingly, Figure 7 illustrates the results of the infection epidemic of each measure with different seed sets for the six networks. As shown in Figure 7, in terms of all of the networks, centroid centrality, which is a shortest-path-based method, performed poorly. One possible reason for this is that centroid centrality, which only takes into consideration the information about the shortest path, does not employ the information that is carried by other paths. Eccentricity centrality and closeness centrality also performed poorly in many datasets, and their underlying principle may explain why. Interestingly, no method did well in all datasets except for EMI, even though stress centrality performed the best in US-Airport. As can be seen in Figure 7, most methods were very unstable and their performance was mixed. Further, the single principle that they followed became ineffective when faced with increasingly complex and dynamic topological characteristics. In US-Airport, E-road, and Chicago, the top 10 most influential nodes as ranked by bridging centrality, centroid centrality, and closeness centrality had only infected dozens of nodes in a network with more than 1000 nodes, which illustrates that these methods had lost effectiveness and mistakenly made some isolated nodes the top key vertices.

Our proposed method EMI was the preferred method among the 11 methods and in the Dutch college, Air traffic control, E-road, Chicago, and Dolphins networks. However, stress centrality and some other methods had slightly better performance in seed sets 10, 20, 30, and 40. The infection epidemic of EMI increased as the seed set increased and caught up with stress centrality in seed set 50. Thus, we could deduce that EMI would perform the best if the seed set grew consistently. In addition, EMI’s infection epidemic in each seed set took first place in the Dutch college, Air traffic control, E-road, and Dolphins networks, which proved that our proposed method, EMI, could precisely identify which nodes were the most important as compared with other famous centrality methods. These datasets basically covered all of the types of complex networks. That is why most of the methods performed exactly the same: sometimes good, sometimes bad. Amazingly, EMI shows relatively stable performance in all datasets. Our proposed method solved this issue once and for all type of complex network.

Finally, the time consumption of each method was considered. We employed Cytoscape, an open source bioinformatics software platform, to run these methods and obtain the uptime. The results of the experiment are shown in Table 7. The proposed method recorded a time of 0.015 s, 0.592 s, 0.039 s, 0.022 s, 0.018 s, and 0.0005 s in Dutch college, US-Airport, Air traffic control, E-road, Chicago, and Dolphins, respectively, which was far less than that of any other method. Therefore, EMI has emerged as the best choice when dealing with large datasets.

## 5. Discussion

Since complex network analysis plays an important role in many fields, complex network science has aroused and will continue to draw sustained attention. To obtain an efficient spread of information, ensure that topological web servers are secure, and optimize the utilization of available resources, the identification of key nodes is of vital significance. In this paper, we proposed a novel entropy and mutual information-based centrality method called EMI. In order to develop a method that not only make full use of local information but also employ all possible characteristics, a wider range of neighbor nodes and all of the digital indexes in the network were taken into account. The algorithm was also shown to remain efficient. In addition, on the basis of repeated observations, we found that some of the well-known centrality methods often overestimate a node’s influence. To overcome this deficiency, several measures were proposed and experimented with. Ultimately, the mutual information-based method emerged as the best candidate. According to [52], mutual information can be used to measure how much duplicate information is there between two information sources. It is worth mentioning that, although the perfect algorithm, free of any limitations and that captures all of the dynamic features, does not exist, EMI still heads the line-up in terms of both the scope of application and the rate of network characteristics usage. EMI can be used in the prevailing types of networks, such as directed networks, weighted networks, and directed and weighted networks. To test the performance of the proposed method, several experiments were done, and we compared EMI with other famous centrality methods, including closeness centrality, betweenness centrality, degree centrality, K-shell, and stress centrality, in six real-world networks (Dutch college, US-Airport, Air traffic control, E-road, Chicago, and Dolphins). The results of the experiments show that EMI had the best performance when compared with the other methods in terms of many evaluation criteria.

Remarkably, there were similarities and differences between EMI and the recently introduced entropy-based method. For example, in [53], the author proposed a subgraph-based approach that takes second-order neighbor nodes only. EMI employs both second-order neighbor nodes and the neighbors of second-order neighbor nodes. Therefore, EMI considers more information when calculating the importance of nodes. However, there remain challenges to be overcome to make our proposed method perfect. Firstly, there remain some ranks that have been assigned to several nodes; that is, we have not yet achieved the goal of an assignment of every node to distinct rank. Secondly, the ability the capture dynamic characteristics needs to improve; for example, there are always users joining and leaving a social network, so the method needs to more fully consider dynamic information for more accurate results.

## Figures and Tables

**Figure 1 entropy-22-00052-f001:**
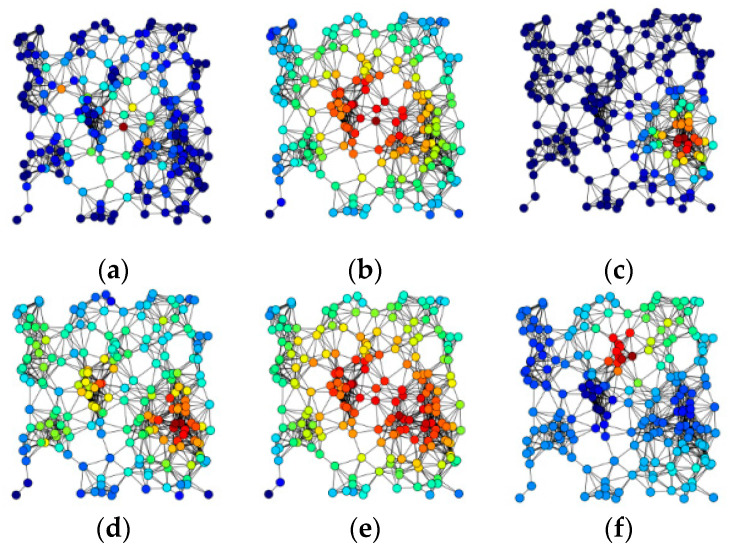
Results of recognizing key nodes in an artificial network by (**a**) betweenness centrality; (**b**) closeness centrality; (**c**) eigenvector centrality; (**d**) degree centrality; (**e**) harmonic centrality; and (**f**) Katz centrality, respectively. These methods are applied in the same artificial network; however, the centralities are quite different from each other (see the red nodes, they are considered as the most important centralities of the network; and the deeper the blue, the less important the node.).

**Figure 2 entropy-22-00052-f002:**
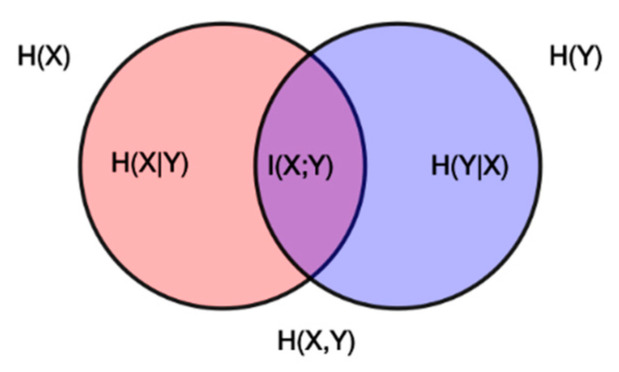
This figure shows the additive and subtractive relationships of various information measures that are associated with correlated variables X and Y. The intermediate part (violet) is the mutual information.

**Figure 3 entropy-22-00052-f003:**
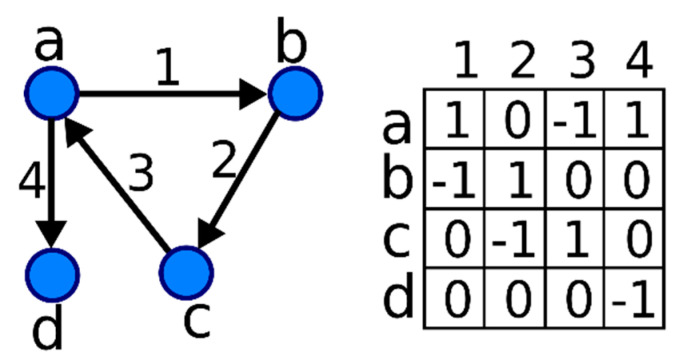
This is a simple directed and weighted network with a corresponding incidence matrix.

**Figure 4 entropy-22-00052-f004:**
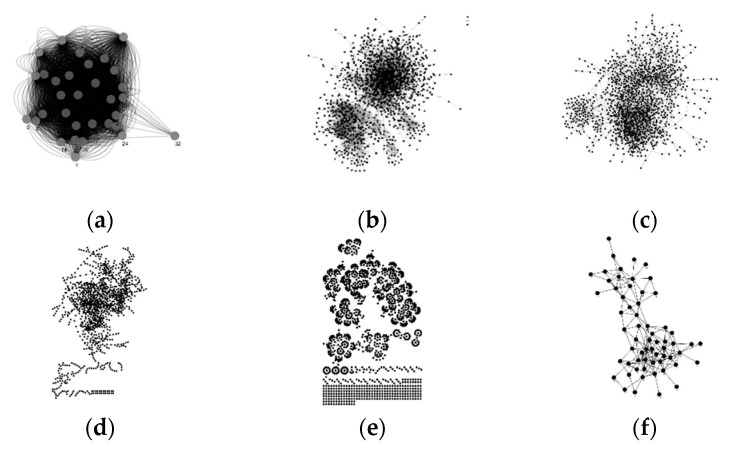
Layout of the datasets. (**a**) Dutch college; (**b**) US-Airports; (**c**) Air traffic control; (**d**) E-road; (**e**) Chicago; and (**f**) Dolphins.

**Figure 5 entropy-22-00052-f005:**
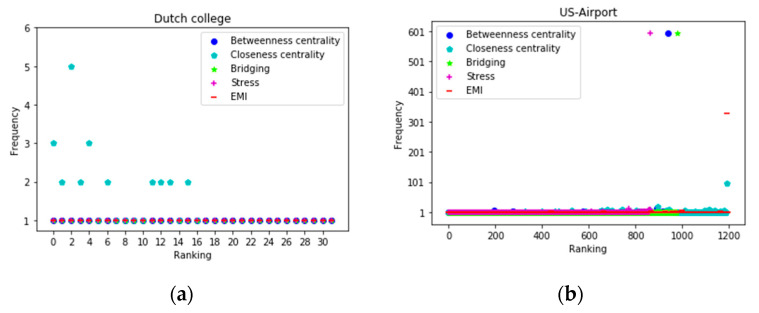
The node distribution as ranked by the top five measures in four datasets: (**a**) Dutch college; (**b**) US-Airport; (**c**) Air traffic control; and (**d**) Dolphins.

**Figure 6 entropy-22-00052-f006:**
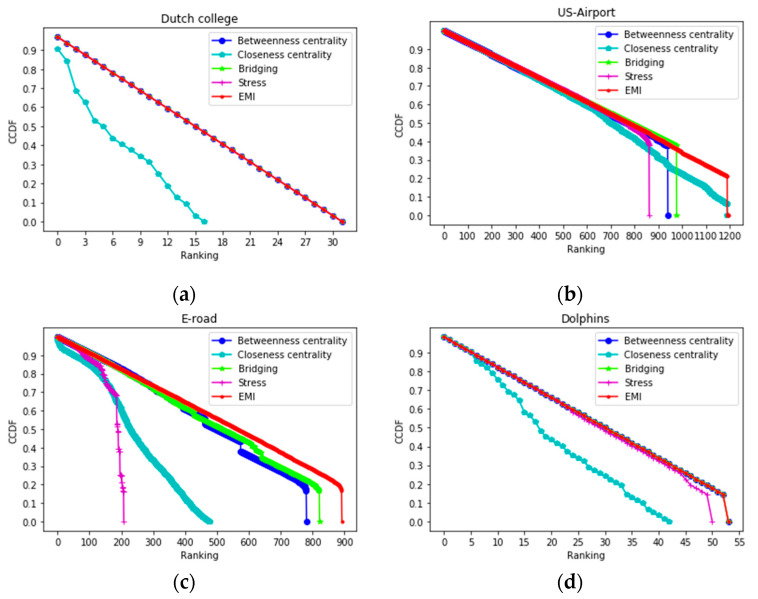
The complementary cumulative distribution function (CCDF) plots for the ranking lists that were obtained by the top five measures on (**a**) Dutch college; (**b**) US-Airport; (**c**) E-road; and (**d**) Dolphins.

**Figure 7 entropy-22-00052-f007:**
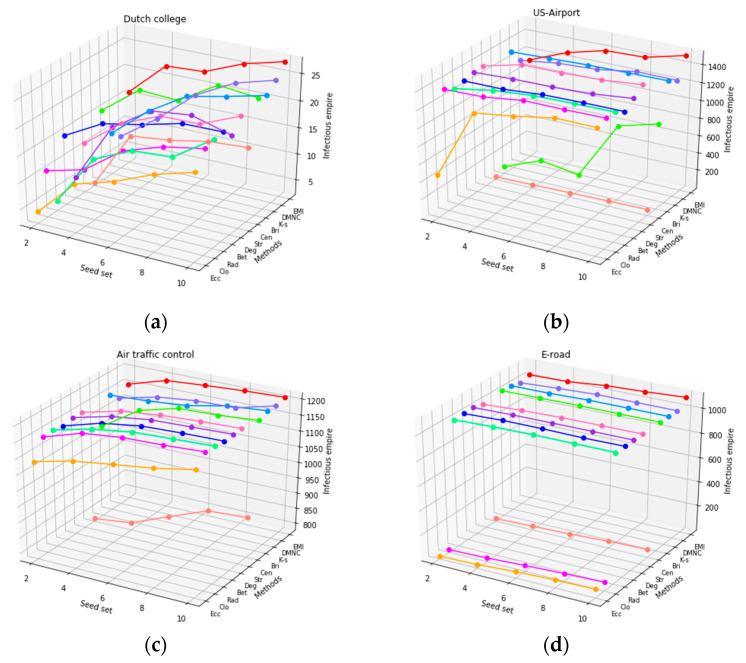
The infection epidemic with different seed sets in the six datasets, including: (**a**) Dutch college; (**b**) US-Airport; (**c**) Air traffic control; (**d**) E-road; (**e**) Chicago; and (**f**) Dolphins.

**Table 1 entropy-22-00052-t001:** This table contains some necessary parameters that can be obtained by Formulas (12), (13), (17), and (18). For example, node *a* and *c* are neighbors of node *b*, and the outdegree of node *a* and *c* is 2 and 1, respectively. Therefore, $EXD_{b}^{out}=3$. The sum of the total weights carried by these outdegrees is 8.

Node	$EXD_{i}^{in}$	$EXD_{i}^{out}$	$EXW_{i}^{in}$	$EXW_{i}^{out}$
*a*	3	2	7	5
*b*	2	3	5	8
*c*	2	3	4	7
*d*	1	2	3	5

**Table 2 entropy-22-00052-t002:** The overall power of each node.

Node	$S_{i}$	$I_{i}$	$MI_{i}$	$K_{i}$
*a*	0.7782	0.7073	−0.03876	1.44674
*b*	0.5775	0.5796	−0.05062	1.10648
*c*	0.5775	0.5040	−0.05062	1.03088
*d*	0.3010	0.2966	0	0.59760

**Table 3 entropy-22-00052-t003:** The ranking list.

Node	Placing
*a*	***NO.1***
*b*	***NO.2***
*c*	***NO.3***
*d*	***NO.4***

**Table 4 entropy-22-00052-t004:** The basic topological properties of six real-world networks.

Network	*n*	*e*	*ad*	*d_max_*	*cc*	ρ	σ
Dc	32	3062	191.38 (overall)	290	0.904	−0.05898	0.981
US-Air	1574	28,236	35.878 (overall)	596	0.384	−0.11330	0.842
A-tc	1226	2615	4.2659 (overall)	37	0.0639	−0.01520	0.951
E-road	1174	1417	2.4140	10	0.0339	0.12668	0.985
Chicago	1467	1298	1.7696	12	0	−0.50492	0.928
Dolphins	62	159	5.1290	12	0.309	−0.04359	0.957

**Table 5 entropy-22-00052-t005:** The M values that were obtained by different methods on the six real networks.

**Dataset**	**M (Rad)**	**M (Cen)**	**M (Ecc)**	**M (DMNC)**	**M (K-s)**	**M (EMI)**
Dc	0.69037	0.00000	0.36785	0.99570	0.12366	**1.00000**
US-Air	0.05925	0.06400	0.00442	0.33589	0.04230	0.85336
A-tc	0.11111	0.03290	0.14406	0.56714	0.86431	**0.94789**
E-road	0.96893	0.40198	0.43244	0.00777	0.81910	**0.97835**
Chicago	0.61174	0.38272	0.09452	0.00000	0.53828	0.83838
Dolphins	0.94540	0.74809	0.11111	0.17084	0.54066	**0.95032**
**Mean value**	**0.56447**	**0.27162**	**0.19240**	**0.34622**	**0.48805**	**0.92805**
**Dataset**	**M (Deg)**	**M (Str)**	**M (Clo)**	**M (Bet)**	**M (Bri)**	**M (EMI)**
Dc	0.97895	1.00000	0.69037	1.00000	1.00000	**1.00000**
US-Air	0.26639	0.27627	**0.98480**	0.36178	0.39860	0.85336
A-tc	0.75375	0.93115	0.90503	0.92784	0.93413	**0.94789**
E-road	0.53086	0.90656	0.96976	0.86180	0.88559	**0.97835**
Chicago	0.07802	0.06905	0.69762	0.69650	**0.93045**	0.83838
Dolphins	0.34183	0.93976	0.94540	0.95030	0.95031	**0.95032**
**Mean value**	**0.49163**	**0.68713**	**0.86550**	**0.79970**	**0.84985**	**0.92805**

The maximum value and mean value are highlighted in bold.

**Table 6 entropy-22-00052-t006:** The top 10 key nodes as ranked by different methods.

Rank	Deg	K-s	Clo	Bet	Str	Rad	Bri	Cen	Ecc	DMNC	EMI
1	22	19	22	7	22	26	7	-	-	**30**	4
2	17	7	3	3	7	22	3	-	-	**11**	5
3	3	3	26	22	3	3	**4**	-	-	18	1
4	7	26	31	6	31	31	**1**	-	-	9	32
5	13	6	20	26	26	20	26	-	-	**10**	24
6	6	31	17	31	21	29	6	-	-	2	30
7	31	22	6	21	13	21	31	-	-	8	11
8	21	21	21	13	29	17	22	-	-	16	25
9	29	29	29	29	6	15	21	-	-	19	10
10	28	13	15	27	17	6	27	-	-	12	14
**Dutch college**											
1	46	-	68	32	22	-	424	-	-	168	99
2	88	-	52	22	10	-	308	-	-	**187**	620
3	69	-	174	165	195	-	498	-	-	169	374
4	74	-	147	74	165	-	1418	-	-	127	187
5	165	-	69	174	147	-	986	-	-	166	265
6	150	-	88	46	174	-	744	-	-	**99**	594
7	174	-	74	418	46	-	804	-	-	**485**	198
8	147	-	159	159	74	-	520	-	-	**265**	485
9	159	-	150	69	317	-	1186	-	-	650	170
10	57	-	60	136	205	-	781	-	-	**620**	665
**US-Airport**											
1	**312**	116	**68**	**68**	**68**	**68**	1226	-	734	3	312
2	**68**	34	**52**	**52**	213	**52**	953	-	369	221	52
3	**52**	**113**	**44**	212	212	**44**	842	-	618	**187**	68
4	**44**	109	**113**	**312**	**52**	**113**	1196	-	678	139	187
5	**187**	102	116	135	220	116	1007	-	827	28	113
6	**113**	77	**47**	213	**312**	**47**	254	-	992	29	44
7	110	424	110	523	135	110	307	-	787	1051	89
8	**47**	81	**46**	220	**44**	**46**	300	-	162	1053	47
9	**89**	308	51	148	148	51	1095	-	464	994	604
10	135	470	**89**	629	119	**89**	396	-	829	841	46
**Air traffic control**											
1	**284**	453	1174	402	277	**401**	404	1174	647	-	284
2	**7**	433	1173	284	402	402	452	1173	59	-	236
3	**137**	542	1163	277	837	403	837	1163	62	-	137
4	**107**	8	1162	453	836	432	835	1163	1174	-	39
5	**236**	179	1152	452	228	1019	801	1151	1173	-	7
6	**39**	280	1151	403	225	253	546	1148	1163	-	107
7	**401**	57	1148	**401**	**284**	452	799	1147	1162	-	401
8	499	253	1147	404	453	404	866	1094	1152	-	43
9	**181**	543	1094	837	452	232	811	1093	1151	-	141
10	**43**	479	1093	836	224	284	889	1077	1148	-	181
**E-road**											
1	**922**	1157	918	**1156**	**1156**	**1156**	1157	918	**922**	-	552
2	**552**	**1154**	798	1157	**1157**	1157	**1156**	798	918	-	922
3	**1156**	**1147**	900	**1147**	**1147**	**1147**	**1147**	900	798	-	1147
4	**1147**	**1156**	1204	**1155**	**1155**	**1150**	**1150**	1204	**552**	-	1150
5	**1155**	**1153**	1163	**1153**	**1153**	**1146**	**1155**	1163	900	-	1146
6	**1153**	**1155**	1085	**1154**	**1154**	**1155**	**1153**	917	1082	-	1153
7	**1154**	**1146**	917	**1150**	**1150**	**1153**	**1154**	1292	917	-	1154
8	**1150**	1134	1292	**1146**	**1146**	**1154**	**1146**	924	559	-	1155
9	**1146**	1148	924	1138	1138	1138	1158	797	1292	-	1156
10	1138	1138	797	499	499	1100	1159	1343	183	-	817
**Chicago**											
1	52	7	**37**	**37**	**37**	**37**	40	**37**	**37**	**30**	21
2	34	**18**	**41**	**2**	**2**	**41**	24	**41**	**2**	**18**	38
3	**18**	22	**38**	**41**	**38**	**38**	8	**38**	**41**	19	15
4	**21**	10	**21**	**38**	**41**	**21**	**37**	**21**	55	25	2
5	58	25	**15**	8	**18**	**15**	29	**15**	8	7	46
6	**30**	**30**	**2**	**18**	**21**	**2**	**2**	8	29	58	41
7	**2**	19	8	**21**	55	29	**41**	34	40	44	18
8	**41**	52	29	55	8	8	55	**2**	31	35	30
9	14	**46**	34	52	58	34	53	9	28	6	51
10	39	**15**	9	58	29	9	33	29	**30**	17	37

Dolphins; Red means that this node is in common with EMI; ‘-’ in the list means this method has neither reliability nor reference value as it has assigned a lot of nodes to one rank.

**Table 7 entropy-22-00052-t007:** The time consumption of different methods in the six datasets. For example, the time of obtaining key nodes of Dutch college using EMI is only 0.0015 s. For each dataset, the shortest uptime is highlighted in bold.

Data	Deg	K-s	Clo	Bet	Str	Rad	Bri	Cen	Ecc	DMNC	EMI
Dc	0.043	0.078	0.055	0.040	0.041	0.042	0.032	0.023	0.025	0.059	**0.015**
US-Air	4.750	4.120	4.520	20.59	4.500	4.390	8.700	4.650	4.580	6.220	**0.592**
A-tc	0.400	0.377	0.413	0.393	0.402	0.412	0.580	0.362	0.400	0.418	**0.039**
E-road	0.495	0.399	0.402	0.413	0.412	0.419	0.501	0.471	0.333	0.413	**0.022**
Chicago	0.520	0.411	0.780	0.790	0.690	0.512	0.742	0.720	0.810	0.991	**0.018**
Dolphins	0.0033	0.0030	0.0043	0.0044	0.0041	0.0037	0.0048	0.0046	0.0050	0.0051	**0.0005**

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
