# Peer review of "Key Node Ranking in Complex Networks: A Novel Entropy and Mutual Information-Based Approach"

_entropy, 2019, doi:10.3390/e22010052_

Round 1

Reviewer 1 Report

This work uses the concept of entropy in order to facilitate the spread of information by choosing key nodes. Albeit the manuscript sounds promising, I see a number of issues that should be resolved before reviewing the work:
- the first part of it (introduction to variety of centrality measures) distracts the reader from from the merits and has no major influence and relevance to the rest
- the manuscript requires significant changes in terms of the quality of writing - some sentences, especially in the introductory part, are meaningless or hard to understand. This is why it is hard to judge the quality of the manuscript so far, since the reader is forced to think too much about what authors wanted to express instead of scientific ideas.

I recommend the authors to resubmit the work with these two issues corrected in order to have it reviewed again.

Author Response

Dear reviewer,

We are very grateful for your insightful comments and suggestions, which helped to significantly improve the quality of this article.

Please see enclosure for details.

Kind regards,

Authors.

Author Response

(The authors gave the same response as above.)

Reviewer 3 Report

Key nodes ranking in complex networks:

The authors propose a novel strategy to rank the nodes in a network. For their purpose, they used information theory-based approaches like Entropy and Mutual information. Although the general idea contains some novelty, the manuscript must be improved properly.  Especially, the language must be improved.

Line 16: we have -> We have intensively

Line 33: [4] has shown -> a paper cannot show it. Please rephrase 

In Figure 1: The color-code

Lines 71 to 74: what about the cumulative Mutual information, like in https://journals.plos.org/ploscompbiol/article?id=10.1371/journal.pcbi.1000978

L 79: Although some excellent .... -> the sentence is not complete

L80 -81: How to take ...-> rephrase it

L105 to 111: section  -> Section (S in capital)

In Formula 1: what is j?

In Formula 2: sum_x is wrong: it must be sum_y

In Formula 3 and 4: C_v(v) -> same variable name is used

L 195: we would like to give a case study to help you well understand EMI -> this sentence is not scientific

L 196 3.1. preliminary -> Preliminary

L200: possible events e_1 ... e_2 -> e_1, ... e_n

L211 it has nothing to do with the hole system. -> ... the whole system

L211-212: Hence, .. -> grammar is not correct

L213 3.2. method description -> Method

L214: where V = {v i , ... , v |V| } -> where V = {v 1, ... , v |V| }

L221: As motivated by what is said in the Introduction -> As motivated  in the Introduction

L226: Only in this way can we -> Only in this way, we  can we

L242: formula -> formulas

L241 to L246: indegree and outdegree entropy centrality S_i^in or S_i^out or S_i  -> do they describe the centrality of the network or individual nodes? It is not very clear

-> Same question is also true  for formula 15 and 16

L247 here are more than structural features -> ...than one structural feature

L270 in figure The -> in Figure the

L272 Caption of Figure 2: Either you should mention what H(X) , H(Y) and H(X,y) mean or remove the Figure. It is not necessary

Formula 18: It is not clear how to call this formula as MI. The analogy to classical Mi calculation must be shown. 

Classical mutual information: MI(x;y) =sum p(x,y) log(p(x,y)/p(x)*p(y))

L294: Do not use in the scientific manuscript the word,, So". Please remove all,, so" in your manuscript

L312: case study -> Case

L321: Caption of Table 2: rewrite it. It does not have any meaning. Further, it is not clear how to calculate the values in Table 2 according to Formulas 8,9,13 and 14. Explain it in a proper way

L357: This is a directed ... -> This directed and weighted ..

L375: Nodes are regard ... -> ... are regarded

L388: ... defined as Eq. (32) -> there is not equation with number 32 in the manuscript

L272 -> Mutual thus -> Mutual information

L412: epidemic process finished -> ... process is finished 

L425: time efficient -> ... efficiency, 

L434: have poor performing -> ... performance

L451: As shown in figure 5(c) -> .. in Figure 5

Figure 5: same methods must have the same symbol in all figures. Further in the figure caption (a), (b) and (c) are bold why not (d)? The same problem exists for other figures as well

L476: key nodes producing by -> ... nodes produced...

L486: Table7: -> The caption must be extended clearly and the color code is missing. 

L497: Last but not list -> ... not least

L559: EMI has the best performance comparing -> ... compared ..  

L575: Acknowledgments: -> Acknowledgements 

General comments:

1) The language must be improved significantly, 

2) The python script or a R-package could be very useful for me to test the method, which is currently missing

3) You should pay more attention to present your method. It must be improved 

Author Response

(The authors gave the same response as above.)

Reviewer 4 Report

In my opinion, this is a good paper, suited to the Entropy journal. I propose to accept it, with the following observations:

Formula 1. If j is v, put tiv in the sum. You say that "j stands for the destination node". If so, then you compute the indegree centrality. Formula 3. What if σst is zero? Formula 22. Specify that you compute the mutual influence of node vi. The sums take x and y, but it is not clear who is j and what is the impact of x and y. The classical model SI also includes R (Recovered, Immune). Please present the theoretical approach. Please review the English. There are several mistakes. For example: vertexes -> vertices.

Author Response

Dear reviewer:

Thank you for your insightful comments, suggestions and your elaborate efforts.

We have fully accepted your suggestions and made the corresponding correction.

Please check the attachment for detail.

Your efforts complete this paper, thank you again!

Authors.

Round 2

Reviewer 1 Report

This version of the manuscript has been significantly improved compared to the previous one. I do have some minor things to ask you to correct in the final version, see below. Still, please do consider whether introducing the SNA measures such as centralities should be so detailed instead of pointing the reader to related works.

Table 7 - the caption should be extended by "expressed in seconds" or similar descriptor of the unit, since only in text we are being told what are the units of the values in the table.

References 44-48 should refer to "KONECT" dataset repository, not "Knoect"

Please consider improving the quality of figures, especially Figure 7

Author Response

(The authors gave the same response as above.)

Reviewer 3 Report

-

Author Response

Dear reviewer:

Thank you for your insightful comments, suggestions and your elaborate efforts.

We have made the corresponding correction.

Please check the attachment for detail.

Your efforts complete this paper, thank you again!

Authors.
